# System identification of neural systems: If we got it right, would we know?

## Abstract

Various artificial neural networks developed by engineers are now proposed as models of parts of the brain, such as the ventral stream in the primate visual cortex. After being trained on large datasets, the network activations are compared to recordings of biological neurons. A key question is how much the ability of predicting neural responses actually tells us. In particular, do these functional tests about neurons activation allow us to distinguish between different model architectures? We benchmark existing techniques to correctly identify a model by replacing the brain recordings with recordings from a known ground truth neural network, using the most common identification methods. Even in the setting where the correct model is among the candidates, we find that system identification performance is quite variable, depending significantly on factors independent of the ground truth architecture, such as scoring function and dataset. In addition, we show limitations of the current approaches in identifying higher-level architectural motif, such as convolution and attention.

## 1 Introduction

The dominant approach for machine learning engineers in search of better models has been to use standard benchmarks to rank model performance. This practice has driven much of the progress in the machine learning community. A standard comparison benchmark enables the broad validation of successful ideas. Recently such benchmarks have found their way into neuroscience with the advent of frameworks like Brain-Score [13], and Algonauts [2], where artificial models compete to predict recordings from brains. Can engineering approaches like these be helpful in the natural sciences?

While such absolute rankings may be a good measure of absolute performance in approximating the neural responses, it is, however, an open question whether they are sufficient to validate or falsify scientific hypotheses in neuroscience. For instance, one of the central questions in neuroscience is about the connnections of neurons and their computational abstraction. In this regard, could the functional similarity imply by itself architectrual similarity? Consider the conjecture that similarity of responses between model units and brain neurons may allow us to conclude that brain activity fits better, for instance, a convolutional motif rather than a dense architecture. If this were actually true, it would mean that functional similarity effectively also constrains architecture. Then the need for a separate test of the model at the level of anatomy would become, at least in part, less critical for model validation.

We describe here an attempt to benchmark the most popular similarity measures by replacing the brain recordings with data generated by a variety of specific known networks, with drastically different architectural motifs, such as convolution vs. attention, thus providing a hopefully useful groundtruth. We also discuss factors that contribute to improving architectural identifiability.

Submitted to 4th Workshop on Shared Visual Representations in Human and Machine Visual Intelligence (SVRHM) at NeurIPS 2022. Do not distribute.

## 2 Background and Methods

### 2.1 Similarity Measures

The two predominant approaches to evaluating computational models of the brain are using metrics based on single-unit response predictivity and population-level representational similarity. Consistent with the typical approaches, we study the following neural predictivity scores: Linear Regression and Centered Kernel Alignment (CKA).

In computational neuroscience, we usually have a neural system (brain) that we are interested in modeling. We call this network a *target* and the proposed candidate model a *source*. Formally, for a layer with $p_1$ units in a source model, let $X \in \mathbb{R}^{n \times p_1}$ be the matrix of representations with $p_1$ features over $n$ stimulus images. Similarly, let $Y \in \mathbb{R}^{n \times p_2}$ be a matrix of representations with $p_2$ features of the target model (or layer) on the same $n$ stimulus images.

**Linear Regression** Closely following the procedure developed by previous works [13, 15, 3], we linearly project the feature space of a single layer in a source model to map onto a single unit in a target model (a column of $Y$). The linear regression score is the Pearson's correlation $r(\cdot, \cdot)$ coefficient between the predicted responses of a source model and the ground-truth target responses to a set of stimulus images. We use ridge regressions with the regularization parameter $\lambda = 1$ for our main experiments and we show the effect of varying the value in Appendix (Figure 6).

$$\hat{\beta} = \mathrm{argmin}_{\beta} ||Y - XS\beta||_F^2 + \lambda ||\beta||_F^2 \tag{1}$$

$$LR(X, Y) = r(XS\hat{\beta}, Y) \tag{2}$$

To reduce computational costs without sacrificing predictivity, we apply sparse random projection $S \in \mathbb{R}^{p_1 \times q_1}$ for $q_1 << p_1$, on the activations of the source model [3]. We use 90% of the stimulus images for linear fitting and test on 10%, cross-validated 10 times. We randomly subsample 3000 units for each target layer and use the median of them as the aggregate score.

**Centered Kernel Alignment** Another widely used type of metric builds upon the idea of measuring the representational similarity between the activations of two neural networks for each pair of images. While variants of this metric abound, including RSA or re-weighted RSA [10, 8], we use CKA [4] as [9] showed strong correspondence between layers of models trained with different initializations, which we will further discuss as a validity test we perform. Recent work [6] notes that under certain conditions linear CKA is equivalent to a whitened representational dissimilarity matrix (RDM) in RSA. We consider linear CKA in this work:

$$\mathrm{CKA}(X, Y) = \frac{||Y^T X||_F^2}{||X^T X||_F ||Y^T Y||_F} \tag{3}$$

### 2.2 Identifiability Index

To quantify how selective neural predictivity scores are when a source matches the target architecture compared to when the architecture differs between source and target networks, we define an identifiability index as:

$$\text{Identifiability Index} = \frac{\text{Score(source = target)} - \text{Mean Score(source} \neq \text{target)}}{\text{Score(source = target)} + \text{Mean Score(source} \neq \text{target)}} \tag{4}$$

### 2.3 Simulated Environment

If a target network is a brain, it is essentially a black box, making it challenging to understand the properties or limitations of the comparison metrics. Therefore, we instead use artificial neural networks of our choice as targets for our experiments. We investigate the reliability of a metric to compare models, mainly to discriminate the underlying computations specified by the model's architecture.

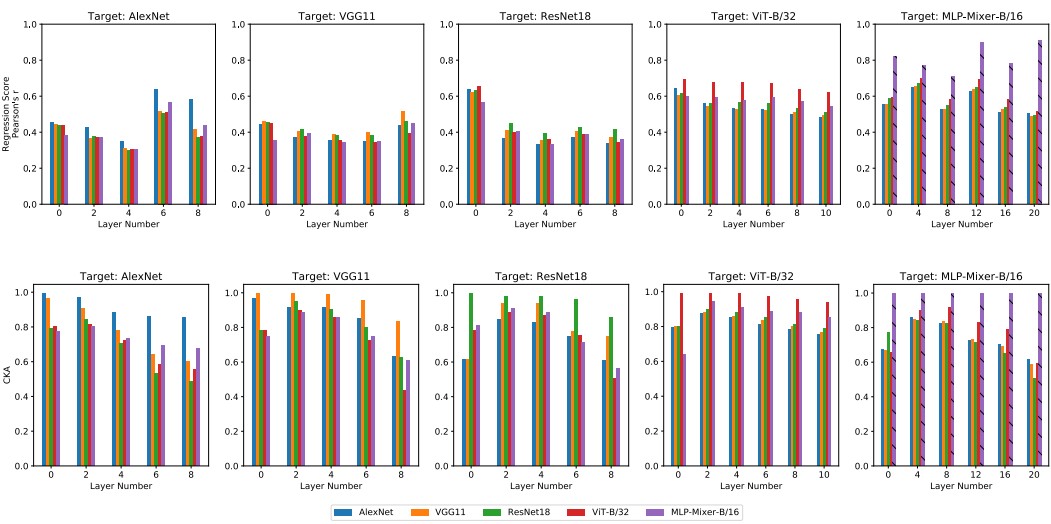

Figure 2: Linear regression (**Top**) and CKA (**Bottom**) scores for artificial neural networks. We use different initialization seeds for source networks of the same architecture type as the target, except for MLP-Mixer-B/16 (bar plots with patterns), for which we test identical weights, the most ideal setting.

## 3 Results

### 3.1 Different models trained on a large-scale dataset reach equivalent neural predictivity

We compare various neural networks based on different components, such as convolutional layers, attention layers, and skip connections as the models of the brain via the Brain-Score framework [13]. Our experiments show that the differences between markedly different neural network architectures are minimal after training (Figure 1), consistent with the previous work [13, 11, 3]. The performance difference is minimal, with the range of scores having a standard deviation $< 0.03$ (for V2=0.021, V4=0.023, IT=0.016) except for V1. For V1, VOneNets [5], which explicitly build in properties observed from experimental works in neuroscience, significantly outperform other models. This suggests that architectures with different computational operations reach almost equivalent performance after training on the same large-scale dataset, i.e., ImageNet.

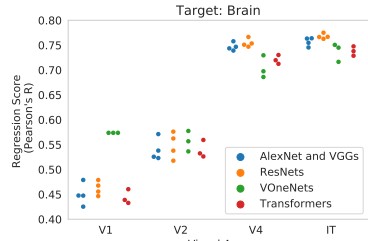

Figure 1: Linear regression scores of deep neural networks for brain activations in the macaque visual cortex.

### 3.2 Identification of architectures in an ideal setting

One interpretation of the result would be that different architectures are indeed equally good (or bad) models of the visual cortex. An alternative explanation would be that the method we use to compare models has limitations in identifying the precise computational operation. To test the hypothesis, we consider the case where underlying target neural networks are known instead of being a black box as with biological brains.

**Linear Regression** We first compare various source models with a target network, the same architecture as one of the source models and is trained on the same dataset but initialized with a different seed. We use images of synthetic objects [12] to be consistent with the evaluation pipeline of Brain-Score. For most target layers, except for those in VGG11, source layers with the highest score are layers in the same network type (Figure 2 top). However, strikingly, for early and intermediate layers of target VGG11, the best-matched layers belong to a source model that is not VGG11. The first layer of ResNet18 is also predicted best by ViT-B/32. In other words, given the activations of VGG11,

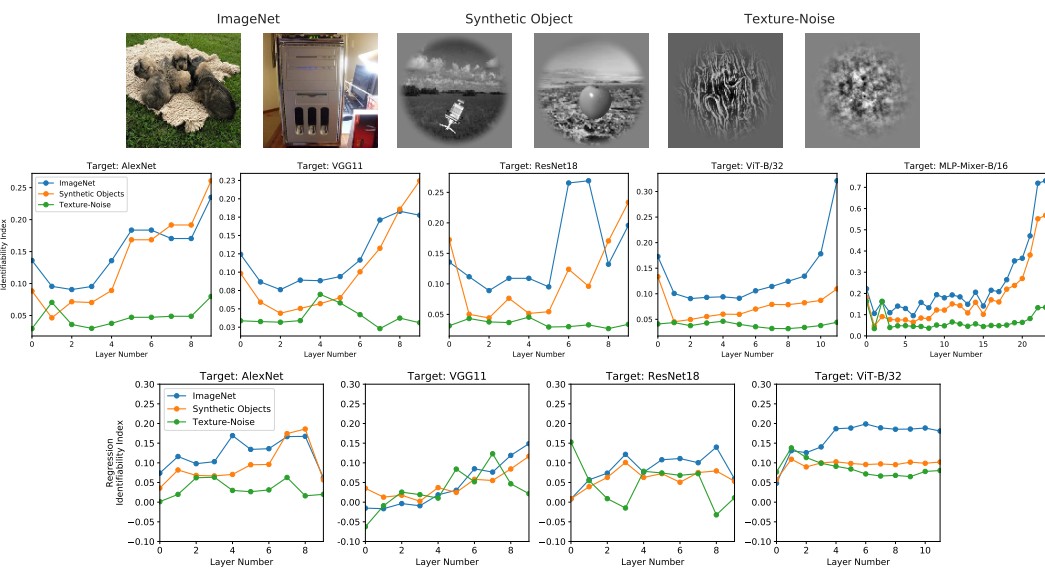

Figure 3: **Top** Sample images of each stimulus image type. **Bottom Two Rows** Architectural identifiability index using CKA and regression for different types of stimulus images.

for instance, and based on linear regression scores, we would make an incorrect prediction that the system's underlying architecture is closest to a ResNet18.

In addition, because of our ideal setting, where an identical network is one of the source models, we expect to see a significant difference between matching and non-matching models. However, for some target layers in AlexNet and ResNet18, although the layer with the highest score may be the matching layer in the same architecture, linear regression scores for other source models do not show a significant decrease in predictivity.

**CKA** Next, we replace linear regression with CKA for the similarity measure. For all layers of the target models, the ground-truth source models achieve the highest score with a significant margin (Figure 2 bottom). To examine how robust CKA is when only a subset of target neurons are available, as with the neural recordings of biological brains, we also test including 1% of target units. We show that the correct source model can still be identified for most layers, but start to observe some layers that are either incorrect or have similar scores across models (Figure 5). Overall, the degree of identifiability decreases, and we expect settings that are more consistent with biology will have even more constraints and noise.

### 3.3 Effects of the stimulus distribution on identifiability

A potentially significant variable in comparing models of the brain is the type of stimulus images. What types of stimulus images are suited for evaluating competing models? In Brain-Score, stimulus images for comparing models of the high-level visual areas, V4 and IT, are images of synthetic objects [12]. In contrast, those for the lower visual areas, V1 and V2, are images of texture and noise [7]. To examine the effect of using different stimulus images, we test images of synthetic objects (3200 images), texture and noise (135 images), and ImageNet (3000 images), which are also the training dataset for models.

In Figure 3, we analyze Identifiability Index for different stimulus images. More natural stimulus images (i.e., synthetic objects and ImageNet) show higher identifiability than texture and noise images. Notably, even for early layers in target models, which would correspond to V1 and V2 in the visual cortex, texture and noise images fail to give higher identifiability.

### 3.4 Challenges of identifying key architectural motifs

Hypotheses for a more biologically plausible design principle of models often involve key high-level architectural motifs. For instance, whether recurrent connections are crucial in visual processing

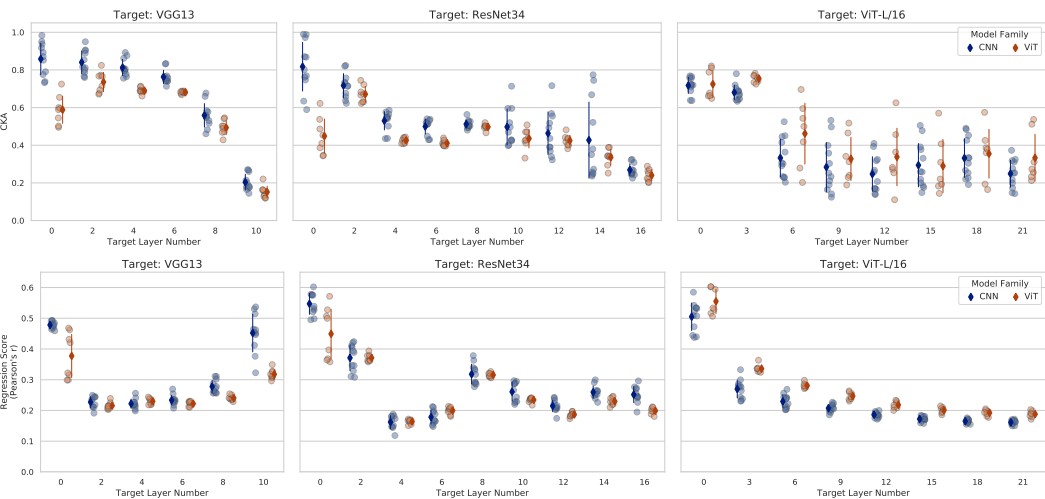

Figure 4: CNNs and ViTs of different architectural variants are compared with two CNNs and a ViT target networks. Each datapoint is the maximum score of an architecture for corresponding target layers. Markers with darker shades indicate mean score of the corresponding model class, and error bars are standard deviation.

or whether the brain implements computations like attention layers in transformers. The details beyond the key motif may vary, and it is unlikely that models align with the brain at every level, from low-level specifics to high-level computation. Thus, an ideal method for comparing models should help separate the key properties of interest while being invariant to other confounds.

Considering it is a timely question, with the increased interests in transformers as models of the brain in different domains [14, 1], we focus on the problem of identifying convolution vs. attention. We test 12 Convolutional Neural Networks and 8 Vision Transformers of different architectures, and to maximize identifiability, we use ImageNet stimulus images. For CKA, we include 1% of target units. Overall, Figure 4 shows that mean CKA and regression scores are higher when target and source models belong to the same model class. However, several layers do not show statistically significant difference between the two model classes based on Welch's t-test with $p < 0.01$ used as a threshold (for CKA, layer 8 of VGG13, layers 2 and 8-16 of ResNet34, and layers 0 and 6-21 of ViT-L/16; for regression, layers 2-6 of VGG13, layers 0-10 of ResNet34, and layer 0 of ViT-L/16).

The significant variance among source models suggests that model class identification can be incorrect depending on the precise variation we choose, especially if we rely on a limited set of models. A quick but essential remedy for this issue is to include wide-ranging variants of a model class rather than to test a single model before concluding high-level key computations.

## 4 Discussion

Under idealized settings, we tested the identifiability of various artificial neural networks with differing architectures. We present two contrasting interpretations of model identifiability based on our results, one optimistic (glass half full) and one pessimistic (glass half empty).

**Glass half full:** Despite the many factors that can lead to variable scores, both linear regression and CKA give reasonable identification capability under unrealistically ideal conditions, with identifiability improving as a function of depth. We find CKA has slightly better reliability than linear regression under these ideal conditions.

**Glass half empty:** However, system identification is highly variable and dependent on the properties of the target architecture and the stimulus data used to probe the candidate models. For architecture-wide motifs, like convolution vs attention, there is significant overlap in scores across almost all layers.

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

# A  Appendix

## A.1  Model details for Section 4.1: Brain-Score

Below is the full list of models tested on the benchmarks of Brain-Score as reported in Section 4.1. In addition to testing vision models pre-trained on ImageNet available from PyTorch's torchvision model package version 0.12, we test VOneNets that are pre-trained on ImageNet and made publicly available by the authors [5]. VOneNets are also a family of CNNs.

**Convolutional Networks:** AlexNet, VGG11, VGG13, VGG19, ResNet18, ResNet34, ResNet50, ResNet101, VOneAlexNet, VOneResNet50, VOnetCORnet-S

**Transformer Networks:** ViT-B/16, ViT-B/32, ViT-L/16, ViT-L/32

## A.2  Model details for Section 4.5: finding the key architectural motif

For each target network reported in Section 4.5, namely VGG13, ResNet34, and ViT-L/16, below is the full list of source models tested to compare two model classes, CNN and transformer. For Tokens-to-token ViTs (T2T) [16], we use models pre-trained on ImageNet and released by the authors. All other models are also pre-trained on ImageNet, available from PyTorch's torchvision model package version 0.12.

**Convolutional Networks**: AlexNet, VGG11, VGG13, VGG16, VGG13_bn, ResNet18, ResNet34, ResNet50, Wide-ResNet50_2, SqueezeNet1_0, Densenet121, MobileNet_v2

**Transfomer Networks:** ViT-B/16, ViT-B/32, ViT-L/16, ViT-L/32, T2T-ViT_t-14, T2T-ViT_t-19, T2T-ViT-7, T2T-ViT-10

## A.3  Model details: number of layers included for each model

Table 1

| Model | Number of Layers |
|---|---|
| AlexNet | 10 |
| Densenet121 | 30 |
| MLP-Mixer_B16_224 | 24 |
| Mobilenet_v2 | 14 |
| ResNet18 | 10 |
| ResNet34 | 18 |
| ResNet50 | 18 |
| Squeezenet1_0 | 13 |
| T2T_ViT_10 | 13 |
| T2T_ViT_7 | 10 |
| T2T_ViT_t_14 | 17 |
| T2T_ViT_t_19 | 22 |
| VGG11 | 10 |
| VGG13 | 12 |
| VGG13_BN | 12 |
| VGG16 | 15 |
| ViT_B_16 | 12 |
| ViT_B_32 | 12 |
| ViT_L_16 | 24 |
| ViT_L_32 | 24 |
| Wide_ResNet50_2 | 18 |

 ## A.4 Supplementary to Figure 2

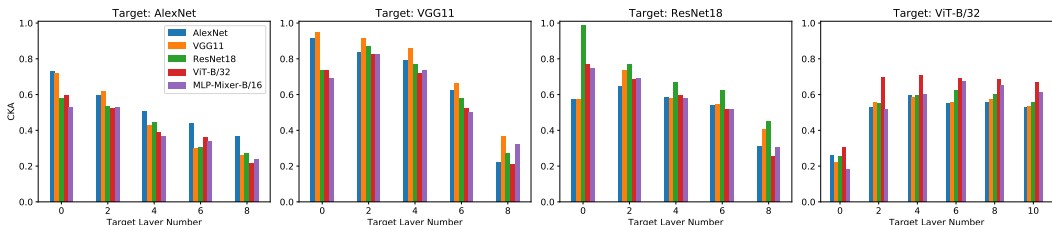

Figure 5: CKA scores when only a subset (1%) of units in a target model are available to be recorded. The constraint is tested to examine whether CKA is reliable in a setting closer to a biological experiment.

 **A.5 Ridge regression regularization coefficient**

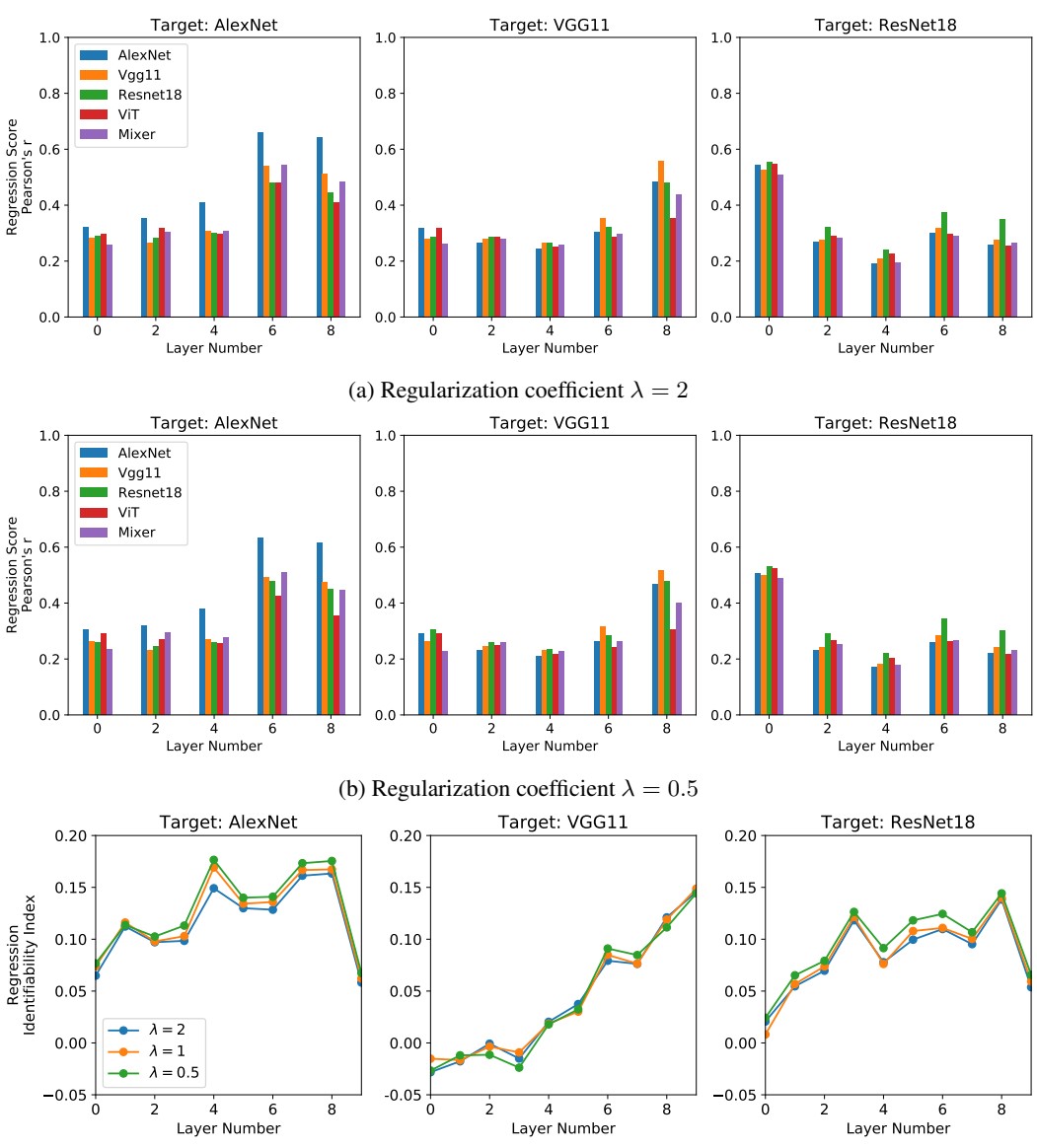

(a) Regularization coefficient $\lambda = 2$

(b) Regularization coefficient $\lambda = 0.5$

(c) Identifiability index for different regularization coefficient values

Figure 6: Results for varying the value of ridge regression regularization coefficient. Stimuli images are from ImageNet.

