# OpenReview forum: "System identification of neural systems: If we got it right, would we know?"
_NeurIPS.cc/2022/Workshop/SVRHM — SVRHM Poster_

### Official Review · Reviewer_dCcW · 2022-10-13
**Important question with some caveats in the implementation**

**Rating:** 6
**Confidence:** 4

**Review:**

The authors ask a simple and fundamental question: Can we use systems identification approaches to identify the true model even when we add the same model as ground truth? The authors conclude that while in many cases it is possible, in some cases the results are quite variable, and that identifying architectural properties (e.g. convolution vs. attention mechanisms) may require the comparison of many networks.

Overall, I enjoyed reading this manuscript. I think that it is in general an important and valuable approach to see whether one can identify ground truth with these approaches.

However, I would like to note some general caveats in the interpretation of the results, as well as some general methodological unclarity that I would like to see clarified:

1. I was surprised how the authors would pull off a comparison of ground truth with the same model. It turns out the authors use merely the same architecture and training data but trained on different random seeds. While it may seem that this would lead to the same model, indeed it has been shown before that there is quite some variability when re-running the same model with different seeds (see https://www.nature.com/articles/s41467-020-19632-w ). Thus, it is to a certain degree questionable that we are dealing with a ground truth model. Ideally, the authors would re-run their analyses with not only a single instance of each model as a target but several instances, as well as some way of norming variability between models (i.e. AlexNet might be more variable in the representation between seeds than VGG in general: how does this affect the results and conclusions drawn by the authors?). I think it would at least be good if the authors discussed this as a limitation of their work and perhaps suggest some alternatives to overcome these limtations.

2. One approach that has been suggested in the past was the method of controversial stimuli (see https://www.pnas.org/doi/full/10.1073/pnas.1912334117 ). I am wondering to what degree this approach would be better suitable for identifying the correct model? Is this something the authors considered? Again, at a minimum I would expect some discussion of this previous work.

3. Another important issue is that it is unclear how this systems identification approach is altered by how neuronal populations or fMRI voxels sample the underlying features. It is possible that this sampling would reduce the ability to fit (e.g. by smearing representations or adding noise) but it is also possible that a combination of several units can lead to better systems identification. Again, I think this is a possible limitation of this approach that would be worth discussing or addressing.

4. It is unclear to me why the L2 regularization parameter was fixed to a lambda of 1. Ideally the authors would run nested cross-validation to determine the optimal parameter. I am not sure how the analysis in the appendix addresses this issue since the range of lambdas is quite limited.

5. CKA is a good evaluation approach but is biased when the diagonal is included. It would make sense to use the modified RV coefficient that excludes the diagonal. See https://arxiv.org/abs/1912.02260

Other comments:
- Lines 55/56: "We randomly subsample 3000 units for each target layer and use the median of them as the aggregate score". Does this mean the authors subsample 3000 units, compute the fit, and use the median correlation coefficient as the aggregate score? Please clarify!
- What is the chosen dimensionality for sparse random projection? And was this chosen based on the Johnson-Lindenstrauss lemma, and if yes, with which epsilon?
- I believe the authors can leave out the formula for linear regression.
- There are numerous typos in the manuscript that could be fixed. Here are some:
* line 6: about neurons -> about neural
* line 14: motif -> motifs
* line 25: conncections -> connections
* line 26: architectrual -> architectual
* line 32: We describe here -> Here we describe
* line 127: we analyze Identifiability Index -> we analyze the Identifiability Index

---

### Official Review · Reviewer_dtS8 · 2022-10-14
**The paper assess the weakness of linear regression and CKA used in relating systems. However, the goals are unclear and the methods might need improvement.**

**Rating:** 6
**Confidence:** 5

**Review:**

This paper shows that linear regression and CKA, methods used to posit equivalence between systems, don't always reveal the expected relationships. The authors consider multiple deep neural networks (DNNs( for their assessments. They show for many of the layers of those DNNs the best match is not the same layer of the same architecture initialised with a different seed. They conclude that using regression and CKA is insufficient to identify what system might be the best model for another system being observed.

The conclusions are fair. However, I am not sure about how non-trivial they are. As also observed by the authors, the representational spaces across DNNs are very similar. When that is the case, is it surprising that sometimes the highest match for a particular layer might come from another model? Furthermore, in the comparisons in Fig. 2, no confidence bounds of the differences are mentioned, and in Fig. 4, while the differences are statistically tested it is unclear that the assessment accounted for multiple comparisons. This additionally makes it hard to place much emphasis on the differences.

In the introduction, the authors ask "could the functional similarity imply by itself architectural similarity"? I do not think that cognitive/systems neuroscientists think this is the case. Are the authors hinting at the need for methods that allow us to disambiguate architectures? If so, that is an interesting question indeed but beyond what regression and CKA have been traditionally used for. It would be useful if the authors could provide suggestions for alternative methods that satisfy their criterion.

Final point related to the authors' suggestion: usually papers studying the relationships between DNNs and neural data do compare multiple DNN architectures before making claims that the representational transformations or the feature dictionaries in DNNs are a good fit for the neural data.

---

### Official Review · Reviewer_gtVn · 2022-10-14
**An important sanity check of some foundational assumptions in comp neuro methods**

**Rating:** 9
**Confidence:** 4

**Review:**

The paper tests a very important question: if we knew which of several visual systems truly produced the neural data in a "benchmark"-style model comparison, would we be able to figure out the correct system, using popular methods like encoding models or RSA?

Pros
+ addresses an important question of interest to most researchers involved with SVRHM
+ presents interesting results, especially in the conclusion that CKA/constrained RSA is generally more effective at identifying the correct system than linear regression encoding models
+ methodologically complex and competent; well justified choices of model and method comparisons, a diverse set of models, and appropriate statistical evaluation

Cons
- no obvious flaws

---

### Official Review · Reviewer_Nf85 · 2022-10-17
**Analysis of important issue with current model-brain comparison studies**

**Rating:** 8
**Confidence:** 3

**Review:**

This paper tests the ability of similarity measures commonly used in model-brain comparison, to predict ground-truth architectural motifs. They determine that a) different-sized models trained on ImageNet achieve near-equal Brain-Scores, b) when the target network, source networks of the same architecture achieve higher similarity scores, c) there is large variance in system identifiability between different stimulus-sets, and d) Model family (Conv or Attention) is difficult to predict from scores.

QUALITY: Clear motivation and description of methods. Simple but well executed idea. Variety of analyses.

CLARITY: Ideas, methods and writing are clear.

ORIGINALITY AND SIGNIFICANCE: Yes. Most recent work comparing deep networks to biological neural activity rely on or are derivatives of similarity metrics defined in this paper. The field has not quite stopped to think about the implications of neural predictivity on system identification of the brain. This paper raises that crucial question and conducts initial, albeit simplistic, experiments which are easily extendable to a deeper analysis.

PROS: 1) Important problem, 2) Variety of useful analyses

CONS: Simplistic environment (assumes homogeneous architecture of target networks, considers only 2 similarity measures)